# Shear Bond Strength and Color Stability of Novel Antibacterial Nanofilled Dental Adhesive Resins

**DOI:** 10.3390/nano13010001

**Published:** 2022-12-20

**Authors:** Qing Hong, Alexandra C. Pierre-Bez, Matheus Kury, Mark E. Curtis, Rochelle D. Hiers, Fernando L. Esteban Florez, John C. Mitchell

**Affiliations:** 1College of Dental Medicine, Midwestern University, Glendale, AZ 85308, USA; 2Division of Operative Dentistry, Department of Restorative Dentistry, Piracicaba School of Dentistry, University of Campinas, Piracicaba 13414-903, Brazil; 3Mewbourne School of Petroleum and Geological Engineering, University of Oklahoma, Norman, OK 73019, USA; 4Division of Dental Biomaterials, Department of Restorative Sciences, College of Dentistry, The University of Oklahoma Health Sciences Center, Oklahoma City, OK 73117, USA

**Keywords:** metaloxide nanoparticles, antibacterial, shear-bond strength, dental materials, human dentin

## Abstract

Experimental adhesives containing co-doped metaloxide nanoparticles were demonstrated to display strong and long-term antibacterial properties against *Streptococcus mutans* biofilms. The present study represents an effort to characterize the shear-bond strength (SBS) and color stability (CS) of these novel biomaterials. Experimental adhesives were obtained by dispersing nitrogen and fluorine co-doped titanium dioxide nanoparticles (NF_TiO_2_, 10%, 20% or 30%, *v*/*v*%) into OptiBond Solo Plus (OPTB). Dentin surfaces were wet-polished (600-Grit). Specimens (n = 5/group) of Tetric EvoCeram were fabricated and bonded using either OPTB or experimental (OPTB + NF_TiO_2_) adhesives. Specimens were stored in water (37 °C) for twenty-four hours (T1), three months (T2), and six months (T3). At T1, T2, or T3, specimens were removed from water storage and were tested for SBS. Disc-shaped specimens (*n* = 10/group; d = 6.0 mm, t = 0.5 mm) of adhesives investigated were fabricated and subjected to thermocycling (10,000 cycles, 5–55 °C, 15 s dwell time). Specimens’ colors were determined with a VITA Easyshade^®^ V spectrophotometer (after every 1000 cycles). SBS data was analyzed using two-way ANOVA and post-hoc Tukey tests, while CS data was analyzed using one-way ANOVA and post-hoc Tukey tests (α = 0.05). Mean values of SBS ranged from 16.39 ± 4.20 MPa (OPTB + 30%NF_TiO_2_) to 19.11 ± 1.11 MPa (OPTB), from 12.99 ± 2.53 MPa (OPTB + 30% NF_TiO_2_) to 14.87 ± 2.02 (OPTB) and from 11.37 ± 1.89 (OPTB + 20% NF_TiO_2_) to 14.19 ± 2.24 (OPTB) after twenty-four hours, three months, and six months of water storage, respectively. Experimental materials had SBS values that were comparable (*p* > 0.05) to those from OPTB independently of nanoparticle concentration or time-point considered. Experimental materials with higher NF_TiO_2_ concentrations had less intense color variations and were more color stable than OPTB even after 10,000 thermocycles. In combination, the results reported have demonstrated that experimental adhesives can establish strong and durable bonds to human dentin while displaying colors that are more stable, thereby suggesting that the antibacterial nanotechnology investigated can withstand the harsh conditions within the oral cavity without compromising the esthetic component of dental restorations.

## 1. Introduction

Composite resins and dental adhesive resins are typically used to repair and augment the function and esthetics of mineralized dental tissues. These mercury-free restorative materials display outstanding handling and esthetic properties, have good mechanical and physical properties, and are associated with minimally invasive [1] and ultraconservative restorative techniques [2]. However, despite their widespread acceptance and utilization, polymer-based adhesive restorations were demonstrated to have limited-service lives (5.7 years) [3] and to primarily fail by secondary caries [4]. Incomplete envelopment of collagen fibrils [3], polymerization shrinkage [5], hydrolysis [6], biodegradation (salivary esterases and biofilms) [7], and upregulation of pathogenic biofilms [8] are some of the typical limitations associated with current dental adhesive resins. Previous studies have indicated that polymer degradation byproducts accumulate at the tooth–adhesive interface and increase the virulence of caries-producing bacteria [9], thereby shifting the ecology of biofilms from a state of health into a disease-associated state [10,11]. The long-term accumulation of degradation byproducts at the adhesive interface has been suggested to result in a degradative positive feedback loop that is responsible for the catastrophic failure of the hybrid layer and the short longevity of polymer-based adhesive restorations [12,13,14].

Several approaches have been investigated to improve the clinical performance and longevity of dental adhesive resins. These include the incorporation of fluoride, quaternary ammonium dimethacrylates (QADM), silver (Ag) [15], zinc oxide (ZnO), and titanium dioxide nanoparticles (TiO_2_) [16,17] into commercially available materials [18,19,20,21,22]. Nitrogen-doped TiO_2_ produced via sol-gel (N-TiO_2_) was demonstrated to enhance the strength and in vitro antibacterial properties of composite resins against *Escherichia coli* when used as a filler [23]. Salehi et al. demonstrated that orthodontic brackets coated with N-TiO_2_ prevented the growth of *Streptococcus mutans* over a period of three months, which was considered to be an effective strategy in preventing enamel demineralization during orthodontic therapy [24]. Sodagar et al., while studying the antibacterial effect of TiO_2_ in orthodontic resins, found that experimental materials containing TiO_2_ (1%, 5%, or 10%, wt/wt) promoted significant in vitro microbial reductions against *S. mutans*, *Streptococcus sanguinis*, and *Lactobacillus acidophilus* [25]. The antibacterial efficacy of silver-doped TiO_2_ (Ag_TiO_2_) has also been evaluated [26]. Results reported have indicated that, independent of presentation (in suspension or immobilized), Ag_TiO_2_ was capable of eradicating *S. mutans* planktonic cultures (strain NG8) when exposed to visible light irradiation (1500 lux) [26]. Despite these promising results, strategies previously reported were not capable of reducing the incidence of secondary caries or to extend the service lives of polymer-based bonded restorations, thereby underscoring a critical need for the development and characterization of novel materials displaying long-term and non-leaching antibacterial and biomimetic properties.

Nitrogen-doped titanium dioxide nanoparticles obtained using two-step solvothermal reactions (N_TiO_2_, 6–15 nm) [27] have been recently characterized and incorporated into a fifth generation and commercially available dental adhesive resin (OptiBond Solo Plus, Kerr Corp., Orange, CA, USA; OPTB). Results indicated that N_TiO_2_ were spherical, had smooth surfaces, and were capable of absorbing two times more visible light when compared to undoped TiO_2_ (P25, Degussa, Germany) [27]. Experimental adhesives containing N_TiO_2_ (50, 67, and 80%, *v*/*v*) were demonstrated to have strong antibacterial and biomimetic properties when irradiated with visible light (410 ± 10 nm, 3 h irradiation = 38.75 J/cm^2^, 24 h irradiation = 310.07 J/cm^2^) [27]. These promising findings precipitated the execution of a follow-up study that focused on the characterization of the water sorption, solubility, and cytotoxicity of experimental adhesives containing N_TiO_2_ (25 and 30%, *v*/*v*) [28]. Materials tested were demonstrated to be less soluble, absorb less water, and to have cytotoxicity properties that were comparable (*p* > 0.05) to those of three commercially available and FDA-approved materials (OptiBond Solo Plus (Kerr Corp., Orange, CA, USA), Clearfil SE Protect (Kuraray Noritake Dental, Tokyo, Japan), and Adper Scotchbond (3M ESPE, St. Paul, MN, USA)) [28].

In combination, previous results cited indicate that N_TiO_2_-containing materials could withstand the harsh conditions in the oral cavity and could express strong antibacterial properties at the adhesive interface without adversely impacting the viability of human pulp cells. Esteban Florez et al. [29], while investigating the utilization of simple methods to synthesize, surface-modify, and functionalize co-doped nanoparticles (nitrogen and fluorine (NF_TiO_2_) or nitrogen and silver (NAg_TiO_2_)) into OPTB, have demonstrated that strategies used resulted in the covalent functionalization of nanoparticles, the establishment of smooth interfaces between nanoparticles and polymer chains, and experimental materials displaying polymeric networks with preserved morphology, structure and three-dimensional lamellar distribution, which indicates that experimental materials tested could potentially display good mechanical properties [29]. Despite these encouraging findings, previous studies [30,31] indicated that the incorporation of nanoparticles (NAg = 0.1% and NACP = 10%, 20%, and 30%) into dental adhesive resins results in experimental materials displaying high viscosity and limited ability to flow into dentinal tubules, as denoted by the formation of resin tags that were shorter than those attained with the parental polymer (Scotchbond Multi-Purpose, 3M, St. Paul, MN, USA).

These results are critical because micromechanical interlocking is the primary mechanism of adhesion to enamel and dentin [32], and the longevity of polymer-based dental restorations has been shown to depend on the quality and durability of the hybrid layer. Since masticatory forces are described as a shearing phenomenon [33], previous studies indicated [34] that bond strength should be tested in shear mode to yield results that are acceptable and relevant from the clinical standpoint [33]. Sirisha and Tanikonda, while reviewing the validity of bond strength tests [35,36], indicated, based on previous scientific evidence [37], that it is important to measure the immediate and aged (three months in water storage, 37 °C) bond strength to predict the clinical effectiveness of dental adhesive resins. Studies conducted by Hidari et al. [38] and Takamizawa et al. [39] have found that in vitro bond strength of commercial self-etch adhesive resins to dentin surfaces was unchanged by storage in water (3, 6, and 12 months). Other studies have reported unchanged or increased bond strengths of self-etch adhesives to dentin at six months and one year. A study in primates found that tensile bond strength of a self-etching primer to teeth extracted 24 h, six months, and one year after restoration placement was not significantly different [40]. Additional studies reported decreased dentin bond strength of etch-and-rinse and self-etch adhesives following long-term water storage [41,42]. Conflicting evidence on long-term performance of adhesive resins warrants additional studies to understand bond durability and color stability of novel dental adhesive resins. Therefore, the objectives of the present study were to characterize the shear bond strength (SBS) and the color stability (CS) of novel experimental adhesive resins containing varying concentrations (10%, 20%, and 30%, *v*/*v*%) of antibacterial NF_TiO_2_. Null hypotheses tested were that experimental materials would display values of shear bond strength and color stability (in terms of Δ*E*_ab_ and Δ*E*_00_) that were similar (*p* > 0.05) to those attained with the commercially available dental adhesive resin.

## 2. Materials and Methods

### 2.1. Synthesis of Nanoparticles

NF_TiO_2_ was synthesized in the laboratories of the Division of Dental Biomaterials at the University of Oklahoma Health Sciences Center College of Dentistry using protocols recently published [29,43,44,45]. In a typical reaction, a solution of 1.7 g of titanium butoxide (Aldrich, St. Louis, MO, USA, 97%), 6.8 g of oleylamine (Aldrich, St. Louis, MO, USA, 70%), 7.10 g of oleic acid (Aldrich, St. Louis, MO, USA, 90%), 0.065 g of ammonium fluoride (Alfa Aesar, Tewksbury, MA, USA), and 0.01250 g of tetramethylammonium hydroxide (Aldrich, St. Louis, MO, USA) was prepared and then mixed with a solution containing 13.10 g of ethyl alcohol (Decon Labs, King of Prussia, PA, USA, 200 proof) and 0.40 g of ultrapure water. Each solution was transparent before mixing, but the final solution became clouded due to micelle formation and some hydrolysis. The final solution was placed in a high-pressure reaction vessel (Parr reactor series 4593, Parr Instrument Company, Moline, IL, USA) lined with a boron-silicate glass liner (Parr Instrument Company, Moline, IL, USA), reacted (180 °C, 24 h, 15 psi), and stirred using an external shaft coupled to a propeller blade (280 rpm). Upon cooling, the solution was decanted and washed with anhydrous ethanol (3 × 1 min/wash) to remove extraneous surfactant. NF_TiO_2_ nanoparticles were then stored in ethanol (concentration ≅ 40 mg/mL).

### 2.2. Synthesis of Experimental Adhesives

Experimental dental adhesive resins were formulated by ultrasonically dispersing 10%, 20%, and 30% (*v*/*v*, suspended in ethyl alcohol) of NF_TiO_2_ into OptiBond Solo Plus (OPTB, Kerr Corp., Orange, CA, USA) using a Q700 sonicator (QSonica, Newtown, CT, USA). The rationale for selecting these concentrations was based on previous publications demonstrating that single-doped or co-doped nanoparticles displayed strong initial antibacterial and biomimetic properties with or without visible light irradiation at similar concentrations [27,45,46,47]. Experimental adhesives were then stored (dark conditions, 25 °C) in the original containers provided by the manufacturer until use (5 mL, black bottles, Kerr Corp., Orange, USA).

### 2.3. Fabrication of Specimens

#### 2.3.1. For Shear Bond Strength Test

De-identified and caries-free extracted human molars (*n* = 110) were longitudinally sectioned (diamond saw, Model 650, South Bay Technology Inc., San Clemente, CA, USA) under copious water irrigation to expose dentin surfaces. Each half tooth section was embedded in an acrylic block and wet-polished through 600-grit SiC paper (Buehler, Lake Bluff, IL, USA) before being randomly distributed to groups investigated. Specimens (*n* = 15/group, height = 2.38 mm, diameter = 2.00 mm) of Tetric EvoCeram (Ivoclar Vivadent Inc., Amherst, MA, USA) were fabricated using a mounting jig (Ultradent Products, South Jordan, UT, USA) and were bonded (37% H_3_PO_4_, 15 s; light cured with Bluephase Style, 20 s top irradiation and 10 s/each side after removal of jig) using the proper adhesive in each group. Figure 1 shows bonded specimens prior to shear bond strength testing. Specimens were then individually stored in distilled water (dark conditions, at 37 °C) for twenty-four hours (T1), three months (T2), and six months (T3). An additional set of specimens (*n* = 15/group/time-point) were fabricated using the same procedures previously described but were bonded (following manufacturer instructions) using unaltered and commercially-available OptiBond Solo Plus (Kerr Corp., Orange, CA, USA), and served as the control group in the present study. At specific time-points (T1, T2 or T3), specimens pertaining to each group investigated were removed from water storage and were lightly dried in preparation for SBS testing procedures.

#### 2.3.2. For Color Stability Test

Disc-shaped specimens (*n* = 10/group; d = 6.0 mm, t = 0.5 mm) were manually fabricated in a custom-made metallic mold using the adhesives investigated (unaltered (OPTB) or experimental (OPTB+ NF_TiO_2_)). All specimens were photopolymerized (385–515 nm, 1000 mW/cm^2^, 60 s, VALO, Ultradent Products, Inc.) against glass.

### 2.4. Shear Bond Strength Test

After the completion of each water storage period (T1, T2 and T3, respectively) specimens from each group were individually tested for SBS. Individual specimens were secured using a metal clamp coupled to an Ultradent shear bond strength testing jig (Figure 2) coupled to an ElectroPuls (model E-3000) universal mechanical testing machine (Instron, Norwood, MA, USA). A notched-edge blade was applied parallel to the face of the specimen within 0.25 mm. SBS tests were performed at a crosshead speed of 0.5 mm/min until fracture or when the applied load dropped by 40% or more, which was then identified as the point of failure. Values of SBS (in MPa) were calculated by dividing the peak load at failure by specimens’ surface areas.

### 2.5. Scanning Electron Microscopy

Samples were allowed to dry for approximately 18 hrs at room temperature and then mounted to aluminum SEM stubs using a conductive carbon adhesive tape. A grounding strip of copper tape was applied to connect the top surface of the sample to the Al stub. Samples were coated with Au/Pd in a Denton Desk V sputter coater system to ensure conductivity. Imaging was performed in a Helios Nanolab^TM^ 600 Dualbeam FIB/SEM (FEI, Hillsboro, OR, USA) at 2 kV accelerating voltage and 0.34 nA beam current. The detector used was a solid-state annular backscatter (ABS) detector. FEI MAPS 2.0 software was used to acquire and stitch together individual SEM images, given that the ROI was not completely within the field of view at the lowest magnification of the SEM.

### 2.6. Color Stability Test

Specimens fabricated as described in Section 2.3.2. were tested for color stability using a hand-held reflectance spectrophotometer (VITA Easyshade^®^ V, VITA, Bad Säckingen, Germany). Specimens’ colors were determined immediately after fabrication and after every increment of 1000 thermocycles (5–55 °C, 15 s dwell time) to a total of 10,000 thermocycles. Raw experimental data was then tabulated, and specimens’ color stability was calculated (in terms of Δ*E*_ab_ and Δ*E*_00_) using the CIELAB and CIEDE2000 [48] equations below and following a previously published protocol [49]. The rationale for the selection of thermocycling as the method to test the color stability of materials investigated was based on a previous study [50] that indicated that thermocycling best simulates the typical physical conditions found in the oral environment (in terms of temperature and humidity).
Δ*E*_ab_ = [(L_2_ − L_1_)^2^ + (a_2_ − a_1_)^2^ + (b_2_ − b_1_)^2^]^1/2^

Δ*E*_00_ = [(ΔL′/K_L_S_L_)^2^ + (ΔC′/K_C_S_C_)^2^ + (ΔH′/K_H_S_H_)^2^ + RT × (ΔC′/K_C_S_C_) × (ΔH′/K_H_S_H_)]^1/2^

where H stands for hue and C for chroma.

### 2.7. Statistical Analysis

Mean and standard deviation values of SBS were calculated and were used to determine statistically significant differences among groups investigated using two-way ANOVA and post-hoc Tukey tests (α = 0.05). Color stability data were assessed for normality and homoscedasticity. Mean values that met statistical assumptions were analyzed using one-way ANOVA and post hoc Tukey tests. Statistical analyses in the present study were performed (α = 0.05) using GraphPad Prism version 9.0.0 for Windows (GraphPad Software, San Diego, CA, USA; available at www.graphpad.com accessed on 15 November 2022) and SPSS 23 (IBM).

## 3. Results

Figure 3A–D illustrate the macroscopic aspects of failed interfaces attained using either OPTB (3A), or experimental adhesives containing 10% (3B), 20% (3C) or 30% (3D) of NF_TiO_2_ after six months of water storage (37 °C, dark storage conditions).

Figure 4A–D reveals the microscopic aspect of failed interfaces attained using either OPTB (4A) or experimental adhesives containing 10% (4B), 20% (4C), or 30% (4D) of NF_TiO_2_ after six months of water storage (37 °C, dark storage conditions).

Mean and standard deviation values of SBS from each group investigated were calculated as described in Section 2.4 and are shown in Figure 5. Two-way ANOVA and post-hoc Tukey tests (α = 0.05) were used to determine the significance level of inter- and intra-group differences at each time point (T1, T2 and T3). Mean values of SBS ranged from 16.39 ± 4.20 MPa (OPTB + 30% NF_TiO_2_) to 19.11 ± 1.11 MPa (OPTB), from 12.99 ± 2.53 MPa (OPTB + 30% NF_TiO_2_) to 14.87 ± 2.02 (OPTB), and from 11.37 ± 1.89 (OPTB + 20% NF_TiO_2_) to 14.19 ± 2.24 (OPTB) after twenty-four hours, three months, and six months of water storage, respectively. Even though parameters “time” (*p* < 0.0001) and “adhesive” (*p* = 0.0005) were observed to be significant predictors of response when analyzed individually, the interaction between parameters of interest (time*adhesive) was shown to not be significant (*p* = 0.8536), which indicates that the functionalization of NF_TiO_2_ into OPTB did not adversely impacted OPTB’s ability to establish adhesive interfaces that were strong and durable.

Table 1 illustrates, in terms of intra-group percent change, the temporal variation in SBS mean values experienced by specimens after each water storage period (T2 and T3) relative to the baseline (T1), where it is possible to observe that SBS values significantly (*p* < 0.001) decreased overtime and independently of the adhesive (control or experimental) and nanoparticles’ concentrations (10%, 20%, or 30%) considered.

Table 2 demonstrates the temporal percent change in SBS values relative to the parental polymer (OPTB), where it is possible to observe that after twenty-four hours of water storage, experimental groups containing either 20% or 30% of NF_TiO_2_ displayed variations in SBS values that were either positive (4.5%) or negative (−2.4%), respectively. After three months of water storage, and except for the group containing 10% of NF_TiO_2_ that had a positive variation in SBS values (+3.3%), all other groups had SBS variations that were negative in nature and ranged from −5.5% (OPTB + 30% NF_TiO_2_) to −1.4% (OPTB + 20% NF_TiO_2_). This trend could not be observed after six months of water storage, and all experimental groups were associated with positive variations in SBS values that ranged from 0.4% (OPTB + 30% NF_TiO_2_) to 5.5% (OPTB + 10% NF_TiO_2_).

In combination, these results suggest that the functionalization of NF_TiO_2_ into OPTB resulted in experimental materials that can establish strong and durable bonds to human dentin. In addition, the statistical analysis performed indicated that materials investigated (control or experimental) had similar (time*adhesive; *p* = 0.8131) SBS values and would be appropriate for clinical use, thereby supporting the utilization of the antibacterial nanotechnology investigated in the present study. Figure 6A–C and Table 3 and Table 4 illustrate the results from the color stability analysis (in terms of Δ*E*_00_ and Δ*E*_ab_, respectively), where it is possible to observe that after 1000 thermocycles, experimental adhesives investigated displayed color variations that were smaller than that of the parental polymer and were below the perceptibility threshold, which indicates that experimental materials were more color stable than OPTB independently of the concentration of nanoparticles considered. From 3000 to 10,000 thermocycles, all materials investigated (unaltered or experimental) were observed to display color variations that were higher than the thresholds of perceptibility (PT) and acceptability (AT). The analysis of Δ*E*_00_ obtained from 5000 to 10,000 thermocycles indicated that experimental adhesives containing 30% of NF_TiO_2_ displayed color variation values that were consistently lower when compared to the control group (OPTB). These results have clearly demonstrated that the incorporation of NF_TiO_2_ into OPTB did not alter the optical behavior of OPTB and seemed to downregulate the color variations observed, which further supports the clinical utilization of the antibacterial nanotechnology investigated here.

## 4. Discussion

Recent advancements in the field of nanotechnology allowed the utilization of nanoparticles and nanostructured materials in many health-related fields, including biosensing, cancer treatment, and drug delivery [51]. Metaloxide nanoparticles (TiO_2_, ZnO, Ag_TiO_2_, N_TiO_2_ and NF_TiO_2_) have gained the attention of researchers due to their ability to generate reactive oxygen species (ROS) and their proven strong antimicrobial and biomimetic effects [52]. Nanoparticles have been utilized in orthodontics, prosthodontics, and restorative dentistry with the objective to improve materials’ mechanical properties and reduce the microbial load adjacent or attached to different types of restorations [53]. Previous studies confirmed that the incorporation of metaloxide nanoparticles into dental polymers resulted in experimental materials with improved physical, mechanical, biological, and biocompatibility properties, thereby supporting the continued development and characterization of nanofilled materials [29,54,55]. However, despite these encouraging findings, recent reports have raised concerns regarding the incorporation of nanoparticles into dental adhesive resins, because nanofilled materials displayed high viscosity and limited ability to flow into dentinal tubules [30,31].

The present study represents an effort to characterize the SBS and the CS of experimental adhesive resins containing varying concentrations of NF_TiO_2_. Results reported in Figure 5 have demonstrated that mean values of SBS peaked after 24 h of water immersion and decreased with the evolution of time (at T2 and T3) independently of adhesive (commercial or experimental) and concentration of nanoparticles considered (10%, 20%, or 30%). According to Esteban Florez et al. [28], the functionalization of hydrolysis-resistant nanoparticles (N_TiO_2_) into dental adhesive resins results in materials that are less soluble, absorb less water, and have higher specific gravity, which in theory could decrease the detrimental effects of water on bond strength. Al-Saleh et al. [56], while investigating the influence of metaloxide nanoparticles on the bond strength and viscosity of dental adhesive resins, have demonstrated that materials containing 5% of either TiO_2_ or ZrO_2_ displayed non-Newtonian rheological behaviors, pseudo-plasticity properties, and improved flow, which suggests that nanofilled adhesive resins could establish strong interfaces with the tooth structure. Results reported [56] have also indicated that aging (thermocycling, 10,000 cycles, 5 °C and 55 °C, 5 s dwell time) led to significant reductions in SBS values independently of the material considered. The authors reported that adhesives containing TiO_2_ nanoparticles were shown to produce enhanced hybrid layers when compared to those attained with ZrO_2_-containing adhesives, as denoted by the formation of resin tags that were longer [56]. In combination, the findings cited corroborate the results of the present study and the rationale for the selection of the nanoparticles investigated.

Sun et al. [57], while investigating the effect of varying concentrations of TiO_2_ (in terms of mass fraction, 0.08%, 0.10%, 0.12%, 0.02%, and 0.5%) in adhesive resins demonstrated that experimental materials displayed SBS values that were comparable (*p* > 0.05) to the parental polymer and indicated the presence of concentration thresholds (minimum and maximum) in which TiO_2_ would yield a positive effect on bond strength. In contrast, the results of the present study revealed that all materials investigated displayed comparable (*p* > 0.05) mean values of SBS at twenty-four hours, three months, and six months independently of concentration of nanoparticles, thereby further corroborating that the functionalization of nanoparticles did not adversely impact the SBS of OPTB. Prior research comparing several self-etch adhesives found that OPTB displayed SBS values that were higher and statistically different (*p* < 0.05) when compared to those from AdheSE, Adper Prompt Self-Etch Adhesive, Clearfil SE Bond, and One-Up Bond F [58]. Even though our results revealed that OPTB displayed SBS mean values that were numerically higher than those of experimental materials at each time point (T1, T2 and T3), differences reported were not statistically significant (*p* = 0.8131).

The results of the present study indicated that no differences in SBS were detected between experimental adhesives containing NF_TiO_2_. This was an unexpected behavior, because previous studies [30,31] have indicated that the incorporation of nanoparticles into dental polymers results in materials with higher viscosity and limited ability to establish adequate hybrid layers. According to Ashraf et al. [31], the agglomeration of nanoparticles reduces the potential enhancement of mechanical properties in nanocomposites due to restriction of interfacial area and inadequate dispersion of agglomerates. Esteban Florez et al. [29], while characterizing experimental adhesives containing as-synthesized and surface-modified nanoparticles using cutting-edge scientific technologies including small-angle neutron scattering (SANS) and time-of-flight secondary ion mass spectroscopy (ToF-SIMS), have demonstrated that the functionalization of non-agglomerated nanoparticles did not affect the composition or the structure of the polymeric network as denoted by materials displaying preserved morphology, radius of gyration, and three-dimensional lamellar distribution [29], which further supports the utilization of the antibacterial nanotechnology investigated in the present study.

Color stability results are shown in Figure 6A–C, where it is possible to observe that thermocycling (5–55 °C, 15 s dwell time) adversely impacted the color of all materials investigated. After 1000 thermocycles, variations in color were observed to be below PT and AT for experimental materials containing either 20% or 30% of NF_TiO_2_. From 4000 to 10,000 thermocycles, all materials investigated displayed variations in color that consistently increased above PT and AT. When comparing (Table 3 and Table 4) the color variation of OPTB and OPTB + 30% NF_TiO_2_, it becomes obvious that the functionalization of NF_TiO_2_ into OPTB resulted in materials that displayed the numerical values of color variation (in terms of Δ*E*_ab_ and Δ*E*_00_) that were consistently lower than that of OPTB, thereby suggesting that NF_TiO_2_ functionalization reduced the impact of thermocycling on the color of OPTB. Figure 6C highlights this trend by demonstrating that color variations detected (OPTB + 30% of NF_TiO_2_) were numerically smaller and statistically significant (Δ*E*_ab_: *p* = 0.044, Δ*E*_00_: *p* = 0.014) compared with those from OPTB after 10,000 thermocycles. Šimunović et al. [59], while investigating the color stability of orthodontic dental adhesive resins, indicated that degree of conversion, water sorption, polymer composition, temperature, and humidity impact the color stability of dental biomaterials through a complex and multifactorial process.

Previous studies [60,61] have indicated that continuous temperature challenges, such as the one used in the present study, typically lead to internal stresses and increased water sorption, and suggested, based on previous scientific evidence [62], that 10,000 cycles correspond to one year of clinical service. El-Rashidy et al. [63], while evaluating the effects of two aging protocols (thermocycling (37–57 °C) for 10,000 cycles and storage in either tea or red wine) on the color stability (in terms of Δ*E*_00_) of commercially available composite resins, have demonstrated that all materials investigated (Omnichroma (Tokuyama Dental, Tokyo, Japan) and Filtek Z350 XT (3M ESPE, St. Paul, MN, USA)) displayed color variations that were statistically significant (*p* < 0.001) and considered unacceptable (above PT and AT) after 10,000 cycles. The results published by Štruncová et al. [50] have corroborated the findings of the present study, when the authors demonstrated that the incorporation of metaloxide nanoparticles (NAg, 0.005–0.025 wt%) resulted in experimental materials displaying improved color stability even after 10,000 thermocycles. According to the authors, color changes observed have precipitated from the composition of the polymer composition, the type of initiator-activator system, and the level of filler particles’ silanization. The incorporation of NAg in the concentrations reported [50] was shown to not change the color of materials investigated, which further supports the utilization of metaloxide nanoparticles in dental polymers for restorative applications.

Results reported in the present study have allowed us to fully accept the first part of the null hypothesis tested, that experimental materials with varying concentrations (10%, 20%, and 30%) of NF_TiO_2_ would display SBS values that were comparable (*p* > 0.05) to those of commercially available adhesive resins. The second part of the null hypothesis was rejected, because experimental materials displayed color variations that were numerically smaller and statistically different (Δ*E*_ab_: *p* = 0.044, Δ*E*_00_: *p* = 0.014) than those from OPTB after 10,000 thermocycles. Once fully developed and characterized, materials reported in the present study may hold the promise to increase the service lives of polymer-based restorations and decrease the incidence of secondary caries and the costs of oral health care. Future studies in the field should investigate the long-term (twelve, twenty-four, and thirty-six months) shear bond strength and color stability of experimental materials in challenging conditions [64] that resemble those of the oral cavity to confirm the promising SBS and CS properties of novel experimental materials investigated.

## 5. Conclusions

The present study has demonstrated that co-doped nanoparticles synthesized via one-step solvothermal reactions can be successfully functionalized into a self-etch and commercially available dental adhesive resin. Experimental materials investigated were observed to display shear bond strength values that were comparable to the parental polymer after twenty-four hours, three months, or six months of water storage. Adhesives containing 30% of NF_TiO_2_ were observed to be the most color stable material after 10,000 thermocycles. Experimental materials displayed positive percent changes in SBS values after six months of water storage, which indicates that adhesive interfaces established with experimental materials may become mechanically more stable when compared to the parental polymer. In combination, results reported support the continued development and characterization of the antibacterial nanotechnology investigated here.

## Figures and Tables

**Figure 1 nanomaterials-13-00001-f001:**
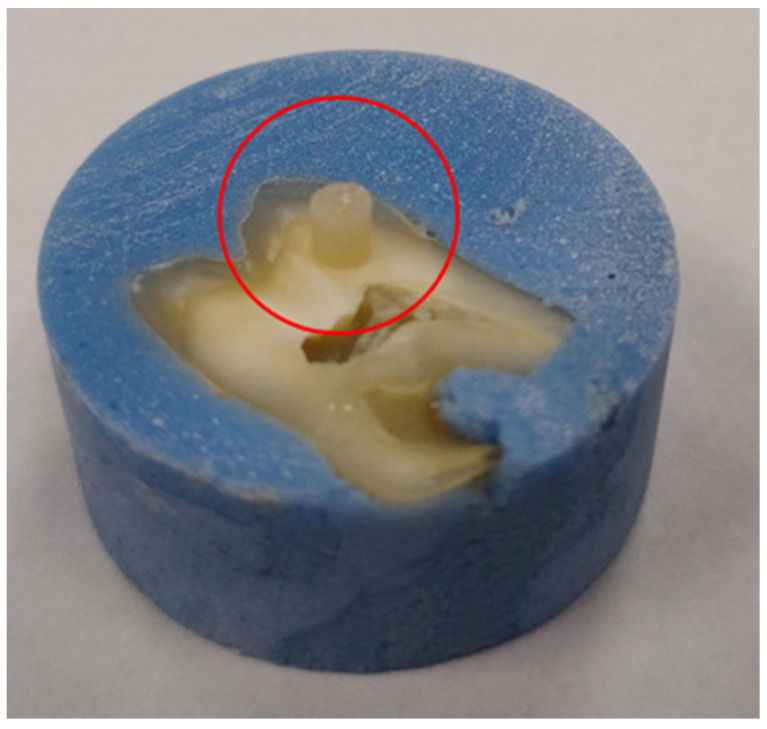
A representative specimen of Tetric EvoCeram composite resin that was bonded using either OPTB, or experimental adhesives containing 10%, 20%, or 30% of NF_TiO_2_. Red circle highlights the location where specimens were typically fabricated.

**Figure 2 nanomaterials-13-00001-f002:**
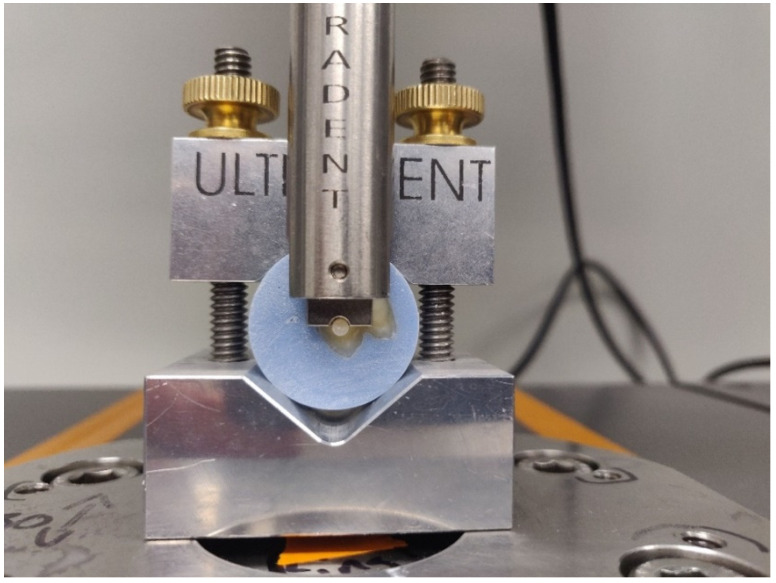
Specimen loaded in the Ultradent shear bond strength testing jig coupled to the E-3000 Instron Universal Testing Machine prior to testing with the notched-edge blade mounted as close to the dentin surface as possible (within 0.25 mm).

**Figure 3 nanomaterials-13-00001-f003:**
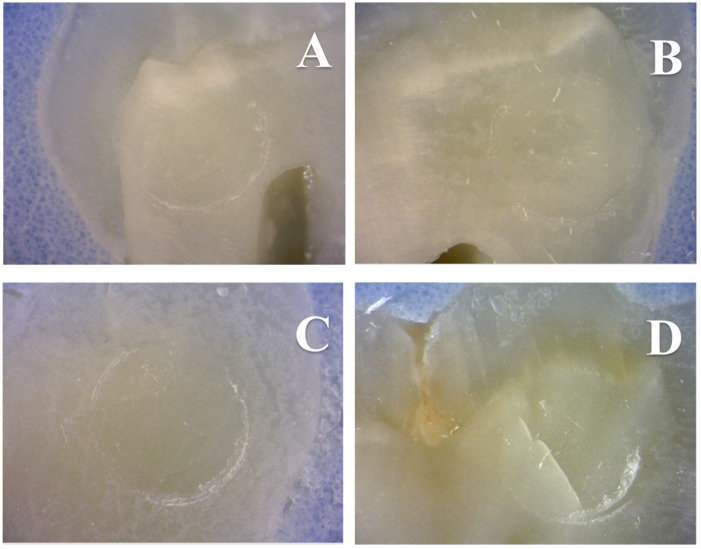
Macroscopic aspects of failed interfaces established either with OPTB (**A**) or experimental adhesives containing 10% (**B**), 20% (**C**), or 30% (**D**) of NF_TiO_2_ and after six months of water storage (37 °C, dark storage conditions). Representative figures indicate that failures were cohesive in resin composite independently of the type of adhesive (unaltered or experimental) or concentration of nanoparticles (10%, 20%, or 30%) considered.

**Figure 4 nanomaterials-13-00001-f004:**
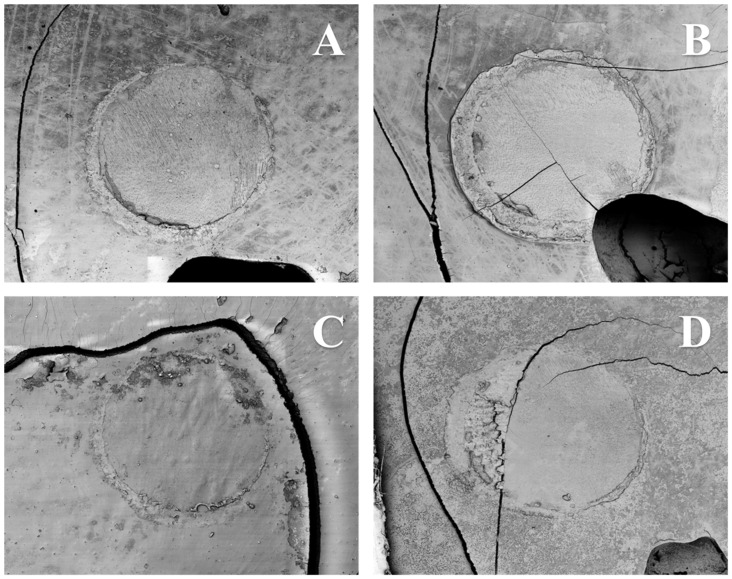
Microscopic aspects of failed interfaces established either with OPTB (**A**) or experimental adhesives containing 10% (**B**), 20% (**C**), or 30% (**D**) of NF_TiO_2_ and after six months of water storage (37 °C, dark storage conditions). Images confirm the findings of Figure 3 and indicate that failures were cohesive in resin composite independently of the type of adhesive (unaltered or experimental) or concentration of nanoparticles (10%, 20%, or 30%) considered. Cracks on images are the result of specimen dehydration, sputter coating and extended periods under vacuum to allow the collection of 25 equally-sized images for the illustration of the entire region of interest at the lowest magnification.

**Figure 5 nanomaterials-13-00001-f005:**
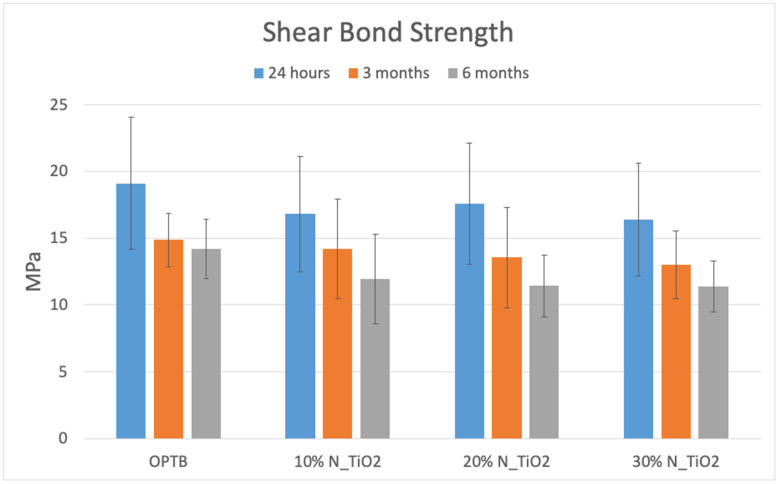
Shear bond strength of unaltered (OPTB) or experimental adhesive resins (OPTB + NF_TiO_2_). Blue, orange, and grey bars indicate 24 h, three months, and six months of water storage (37 °C), respectively.

**Figure 6 nanomaterials-13-00001-f006:**
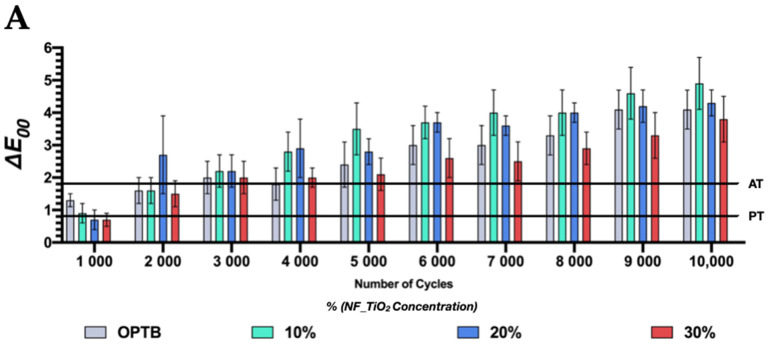
Color stability in terms of (**A**) Δ*E*_00_ and (**B**) Δ*E*_ab_ of unaltered and experimental dental adhesive resins. (**C**) the color stability of materials investigated after 10,000 thermocycles. Dissimilar letters above individual bars denote the presence of statistically significant differences (*p* < 0.05).

**Table 1 nanomaterials-13-00001-t001:** Intra-group percent change in shear bond strength over time.

	24 h–3 Months	3 Months–6 Months
OPTB	−22.1%	−4.6%
10% N_TiO_2_	−15.4%	−16.0%
20% N_TiO_2_	−22.8%	−15.9%
30% N_TiO_2_	−20.7%	−12.5%

**Table 2 nanomaterials-13-00001-t002:** Percent Change—Experimental adhesives from OptiBond Solo Plus.

	24 h	3 Months	6 Months
10% N_TiO_2_	0.0%	3.3%	5.5%
20% N_TiO_2_	4.5%	−1.4%	0.8%
30% N_TiO_2_	−2.4%	−5.5%	0.4%

**Table 3 nanomaterials-13-00001-t003:** Mean and standard deviation values of color variation (in terms of Δ*E*_00_) after each specific thermocycle investigated.

Δ*E*_00_										
GROUP	NUMBER OF CYCLES									
	1000	2000	3000	4000	5000	6000	7000	8000	9000	10,000
OPTB	1.3 (0.2)	1.6 (0.4)	2.0 (0.5)	1.8 (0.5)	2.4 (0.7)	3.0 (0.6)	3.0 (0.6)	3.3 (0.6)	4.1 (0.6)	4.1 (0.6)
10% NF_TiO_2_	0.9 (0.3)	1.6 (0.4)	2.2 (0.5)	2.8 (0.6)	3.5 (0.8)	3.7 (0.5)	4.0 (0.7)	4.0 (0.7)	4.6 (0.8)	4.9 (0.8)
20% NF_TiO_2_	0.7 (0.3)	2.7 (1.2)	2.2 (0.5)	2.9 (0.9)	2.8 (0.4)	3.7 (0.3)	3.6 (0.3)	4.0 (0.3)	4.2 (0.5)	4.3 (0.4)
30% NF_TiO_2_	0.7 (0.2)	1.5 (0.4)	2.0 (0.5)	2.0 (0.3)	2.1 (0.5)	2.6 (0.6)	2.5 (0.6)	2.9 (0.5)	3.3 (0.7)	3.8 (0.7)

**Table 4 nanomaterials-13-00001-t004:** Mean and standard deviation values of color variation (in terms of Δ*E*_ab_) after each specific thermocycle investigated.

Δ*E*_ab_										
GROUP	NUMBER OF CYCLES									
	1000	2000	3000	4000	5000	6000	7000	8000	9000	10,000
OPTB	1.0 (0.2)	1.7 (0.6)	2.3 (0.6)	1.8 (0.6)	2.5 (0.7)	3.2 (0.7)	3.2 (0.7)	3.6 (0.8)	4.5 (0.7)	4.4 (1.1)
10% NF_TiO_2_	1.1 (0.4)	1.7 (0.6)	2.5 (0.8)	3.2 (0.9)	3.9 (0.9)	4.2 (0.6)	4.6 (1.1)	4.8 (1.1)	5.1 (0.9)	5.3 (1.0)
20% NF_TiO_2_	0.9 (0.5)	3.8 (1.8)	2.5 (0.7)	3.5 (1.4)	2.8 (0.5)	4.1 (0.7)	3.9 (0.5)	4.4 (0.5)	4.7 (0.7)	4.6 (0.5)
30% NF_TiO_2_	0.9 (0.5)	1.7 (0.5)	2.2 (1.0)	2.3 (0.5)	2.2 (0.6)	3.0 (0.7)	2.6 (0.7)	3.1 (0.7)	3.5 (0.7)	4.2 (0.6)

## Data Availability

Datasets generated and analyzed in the present study are available from the corresponding author on reasonable request.

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
