# Peer review of "Shear Bond Strength and Color Stability of Novel Antibacterial Nanofilled Dental Adhesive Resins"

_nanomaterials, 2022, doi:10.3390/nano13010001_

Round 1

Reviewer 1 Report

Reviewer's comments:

 In this paper, the long-term shear bond strength (SBS) and the color stability (CS) of novel experimental adhesive sresins containing varying concentrations (10%, 20% and 30%, v/v %) of antibacterial NF_TiO2. Null hypotheses tested were that experimental materials would display values of shear bond strength and color stability (in terms of ΔEab and ΔE00) that were similar (p > 0.05) to those attained with the commercially available dental adhesive resin. However, this article lacks some images of micro morphology and color changes. I suggest that the author provide the micro morphology of the joint surface, and the macro morphology of samples at different times should also be provided.

Reviewer 2 Report

Dear Authors, below are my comments about the submitted manuscript.

1. The title should be reformulated. The term “long-term” cannot represent the clinical scenario since your simulation stops at 6 months. It is not even close to a mid-term in clinical conditions.

2. The abstract section is properly structured but it too long. Please synthetize. Give eminent data in the results of abstract, not only significance.

3. “However, despite their wide-spread acceptance and utilization, polymer-based adhesive restorations were demonstrated to have limited-service lives (5.7 years) and to primarily fail by secondary caries. Incomplete envelopment of collagen fibrils, polymerization shrinkage, hydrolysis, biodegradation (salivary esterases and biofilms) and upregulation of pathogenic biofilms, are some of the typical limitations associated with current dental adhesive resins.” Please add reference to these statements.

4. Pag. 3 line 132: Please reformulate the term “long term”.

5. You should give a better description on how the shear bond strength test was performed, adding a representative figure of specimens mounted on the machine. Moreover, which are your criteria to define fracture?

6. Table 1 and Figure 1 replicate the same results? Choose one where summarize and eliminate the other one.

7. Please provide some images of the samples, before and after testing.
